# Reproducing Improvement-Focused Causal Recourse

## 1 Introduction

With the rapid increase in the usage of algorithms in decision-making processes across various domains, concerns have been voiced upon the potential of producing unfair and biased outcomes. As a result, the concept of algorithmic recourse was introduced as a tool to help understand how the algorithm came upon the decision and possible steps to reverse a decision, in an attempt to reach a more favorable decision. There are two main approaches to algorithmic recourse: Counterfactual Explanations (CE)Wachter et al. (2017) and Causal Reasoning (CR)Karimi et al. (2020). CE aims to explain the decision by displaying different scenarios that would lead to a different and more ideal outcome. On the other hand, CR provides a method for identifying the minimal interventions, which are based on causal relationships in the data, resulting in a more favorable decision. Both approaches provide insight into the decision-making process of the algorithm. However, they work towards modifying the symptoms, in order to 'trick' the model into acceptance, leaving the real-world situation unchanged. To solve this limitation, Improvement-Focused Causal Recourse (ICR) König et al. (2023) was built upon the foundation of CR, to extend its act by suggesting actions to revert the models decision and result in improvements in the real world. The aim of this study is to examine the reproducibility of the original paper by König et al. (2022) and verify its main claims.

## 2 Scope of reproducibility

The original paper had focused upon researching the development of the three algorithmic recourses, through the effectiveness of ICR, intervention stability and cost considerations. Generally, the experiments were able to validate and support their claims in a fair and accountable way. The objective is to reproduce the results from the experiment to prove that the claims hold.

- **Claim 1 :** ICR outperforms CE and CR in terms of improvement.

- **Claim 2 :** ICR obtains the desired acceptance rates.

- **Claim 3 :** ICR acceptance rates remain stable across model refits.

- **Claim 4 :** CE and CR actions are cheaper than ICR.

We extended our research by conducting an extensive investigation into ablation techniques with multiple variables (5var and 7var). A customized synthetic 4var SCM was employed to provide a controlled environment for testing if the authors' claims generalise to other models as well or not. Hyperparameter tuning, focusing on threshold and population size, was implemented to potentially optimise the model's performance.Zhai et al. (2021) Additionally, an extra experiment involved correcting the original 5var dataset, demonstrating a commitment to accurate reproducibility in the future. These extended efforts aimed to enhance the validity of the claims by offering a more nuanced understanding of the models.

## 3 Methodology

### 3.1 Model description

The Improvement-Focused causal models are defined as an explanation technique that targets the improvement instead of the acceptance. For the experiments, the authors describe two different settings, each with its own model definition. These are defined as follows:

**Individualized improvement confidence** In this setting, the model works with the information obtained from a structural causal model (SCM). This is done by estimating what the outcome would have been if the individual had acted differently. Hence, both the prediction and target variables are considered separately.

Based on this, the probability of the individual's acceptance, as well as their probability of improvement is calculated as follows:

$$\gamma^{\text{ind}}(a) = \gamma(a, x^{\text{pre}}) := P(Y^{\text{post}} = 1 | \text{do}(a), x^{\text{pre}})$$

where, $\gamma^{\text{ind}}$ is the individualised improvement confidence of an action $a$ and pre-recourse observation $x^{\text{pre}}$, and $Y^{\text{post}}$ is the target variable post-recourse.

**Subpopulation-based improvement confidence** In this setting, if the SCM is not specified but the causal graph is known instead, and there are no unobserved confounders the effects of the interventions can still be estimated. The population is narrowed down to a sub-population of similar individuals, for which the model can then estimate the sub-population based causal effect.Whitlock et al. (2017) If an action $a$ potentially affects the target Y, then the subpopulation-based improvement confidence $\gamma^{sub}$ is defined as:

$$\gamma^{sub}(a) = \gamma(a, x_{G_a}^{pre}) := P(Y^{post} = 1 | doa, x_{G_a}^{pre})$$

where $G_a$ is the characteristics of the sub-group.

## 3.2 Datasets

The original dataset used by the authors is utilized to reproduce the results of the original paper. The datasets are sourced from existing causal studies by Montandon et al. (2021) and Jehi et al. (2020). Additional pre-processing was unrequired, because the same dataset was utilized. A description of the datasets can be seen in Table 1.

| Dataset | Description | Features used |
| --- | --- | --- |
| 3var-causal | Abstract, synthetic setting | - |
| 3var-noncausal | Abstract, synthetic setting | - |
| 5var-skill | Using GitHub profiles to detect developer role (Montandon et al., 2021) | Senior-level skill (binary), programming experience, education degree, GitHub metrics: commits, languages, stars |
| 7var-covid | Status after covid tests (Jehi et al., 2020) | Population density, flu vaccination, covid vaccination shots, average BMI deviation, covid status, influence on individual, appetite loss, fever, fatigue |

Table 1: Description of datasets used in original paper

## 3.3 Extensions

**Ablation studies on special edges**

A special edge is a link between an exogenous variable and an endogenous effect variable. Ablation studies have been performed by removing these special edges from the causal graphs of the SCMS for 5var-skill and 7var-covid. The causal graphs in Fig 1a display the difference between the removal, where the blue lines indicate the edges that would be removed. By doing this, we try to understand the importance of these edges and their possible effects. Also, we verify if the authors' claims are still consistent even if we remove these special edges. Once we remove the special edge, we also delete the structural equations corresponding to that edge.

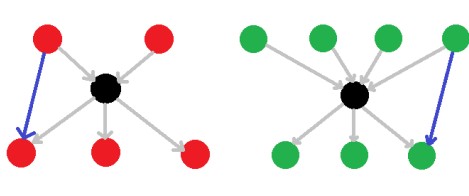

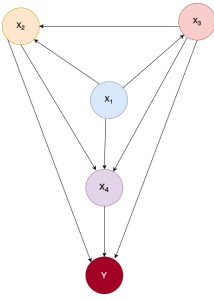

(b) Synthetic 4var causal graph

(a) Ablation edge causal graphs

Figure 1: Causal graphs

**Testing on a different synthetic dataset - 4 var SCM**

Here we created our own 4-var SCM. We want to verify if the authors' claims hold true for this synthesized dataset. The causal graph for our synthesized dataset looks like Fig.1b As we see, $X_1$ is an exogenous variable, where as $X_2$,$X_3$,$X_4$ and $Y$ are exogenous variables. $X_1$ is the causal variable for $X_2$,$X_3$ and $X_4$ and $X_2$, $X_3$ and $X_4$ are the causal variables for $Y$. $Y$ here is the outcome variable. The Structural Equations are as follows:

$$X_1 = U_1, \qquad U_1 \sim N(0,1)$$

$$X_2 = X_1 + X_3 + U_2, \qquad U_2 \sim N(0,1)$$

$$X_3 = X_1 + U_3, \qquad U_3 \sim N(0,1)$$

$$X_4 = X_1 + X_2 + X_3 + U_4, \qquad U_4 \sim N(0,1)$$

$$Y \sim [\sigma(X_2 + X_3 + X_4) <= U_Y], \qquad U_Y \sim Unif(0,1)$$

**Using different mutation methods while performing recourse**

In the original approach proposed by the authors, Gaussian mutation was employed on the population to enhance both the quality of the population and the exploratory nature of the search space. To further refine the algorithm and assess the efficacy of Gaussian mutation, two alternative mutation techniques, namely flip bit mutation and shuffled index mutation, were introduced.

In the case of Shuffled Index Mutation, the attributes of the input individual undergo shuffling to produce a mutant, while in Flip Bit Mutation, the attribute values are flipped to generate a mutant.

**Using different selection methods while performing recourse**

In the original approach, a dual-stage optimization methodology was employed. Initially, the authors utilized Gaussian mutation for exhaustive exploration of the search space. Subsequently, a Non-dominated Sorting Genetic Algorithm-II (NSGA-II)Deb et al. (2002) was applied to select the optimal member within the mutated population, considering the cost function.

The preceding section of our experimentation focused on various mutation methods. In this section, attention is directed towards exploring different selection methods aimed at refining the population towards individuals demonstrating superior performance concerning specified objectives.

Distinct selection methods were employed, including:

*Random Selection*: This method disregards fitness values and randomly selects k individuals from the population. It is valuable in scenarios prioritizing diversity, employing a purely random mechanism.

*Best Selection*: This method employs a straightforward strategy, selecting the top-k individuals with the highest fitness values. It assumes that higher fitness equates to superior individuals, often used in evolutionary algorithms for maintaining high-performing individuals across generations.

*Strength Pareto Evolutionary Algorithm II (SPEA-II) Based Selection*: This approach applies the SPEA2 algorithm for selection, considering both dominance strength and raw fitness. It aims to sustain diversity in the Pareto front, ensuring a well-distributed set of solutions across the trade-off space.

**Changing the threshold parameter ( thres )**

Adaptive threshold is the confidence threshold. It verifies the significance of the changes, so that it effectively provides change that is advantageous. This adaptation was motivated by its potential impact on the likelihood of accepting recommendations and the acceptance to making changes. In order to study the effects of the threshold parameter, we conduct the experiments with different values. The default setting of the threshold value is 50%. We conduct additional experiments for threshold values 20% and 80% and verify if the authors' results are robust and their claims are valid. Other parameters remain unchanged.

**Changing the population size**

Population size is used for the optimization process, thus crucial to tune, because the diversity of the produced solutions is highly influenced by this factor.

As per the default setting, the population size is 300 for the 7-var-covid SCM. As part of this experiment, we double the population size and verify the claims. Also, we verify the robustness and check if there are too many deviations from the authors' reported results.

### 3.4   Computational requirements

All experiments were performed through Snellius[1], using the NVIDIA A100-SXM4-40GB MIG 3g.20gb accelerator. The reproduced experiments cost 4 days to execute, whereas the extended experiments took approximately 13 days. Refer to Table. 2 and Table. 3 to find the detailed running times.

| Dataset | Running Time |
|---|---|
| **5 var Skill** | 1-10:11:41 |
| **7 var Covid** | 1-02:59:52 |
| **3 var Non-causal** | 13:25:28 |
| **3 var Causal** | 12:16:18 |
| **4 var Synthetic** | 17:02:18 |
| **Total Running Time** | 4-14:37:13 |

Table 2: Computational requirements for reproducibility experiments

| Dataset | Running Time |
|---|---|
| **5 var edge deletion** | 1-09:04:58 |
| **7 var edge deletion** | 1-05:05:58 |
| **3 var-nc best select with ear** | 11:58:20 |
| **5 var skill anomaly fix** | 2-03:46:07 |
| **Flip bit __3 var nc** | 12:18:21 |
| **7 var hyper param tuning** | 2-00:00:18 |
| **3 var nc random select** | 12:16:32 |
| **shuffle index 3 var nc** | 12:20:25 |
| **3 var nc SPEA** | 12:15:40 |
| **Threshold 7 var 80 percent** | 2-01:23:12 |
| **Threshold 7 var 20 percent** | 1-03:19:02 |
| **4 var synthetic dataset** | 16:58:01 |
| **Total Running Time** | 13-04:46:54 |

Table 3: Computational requirements for extended experiments

---

[1]A Dutch supercomputer used for large and complex models for universities.

# 4  Results[2]

The reproduced results generally align along with the claims made in the original authors paper. The majority of the extended research results align with the claims of the original authors, which can be seen in Table 4.

## 4.1  Reproduced Results

**Claim 1: ICR outperforms CE and CR in terms of improvement -** *Correct*

ICR outperforms CE and CR in terms of improvement. As depicted in Fig. 2a, ICR was able to produce improvement rates which were all above 0.5. On the other hand, all rates for CE were below 0.5, whereas CR had the 3 var-causal perform slightly better than 0.5. This trend is confirmed by the results produced by the original authors as seen in Fig. 4a. In other words, through the reproduced results, one is able to confirm the validity of claim 1. ICR focuses on improvement rates, whereas CE and CR yield low improvement rates.

**Claim 2: ICR obtains the desired acceptance rates -** *Correct*

Despite the focus on improvement, ICR was able to achieve a decent level of acceptance, allowing the recourse to be of significant value. ICR's focus on improvement, and hence the low acceptance values, along with the relatively high values for CE and CR is illustrated in Fig. 2b. Thus, claim 2 can be proven to be valid as all the methods do yield desired acceptance rates.

**Claim 3: ICR acceptance rates remain stable across model refits -** *Correct*

The acceptance rates for ICR are robust as shown in Fig. 2c, meaning that changes to the datasets do not induce huge deviations in the results. The reproduced results align well while having some minor differences. This could be due to the usage of different random seeds or noise.

**Claim 4: CE and CR actions are cheaper than ICR -** *Correct*

Table 4e clearly displays that the ICR actions are costlier than CE and CR. In our reproduced results, the standard deviations obtained were significantly smaller than the ones obtained by the original author. This is due to the 5var SCM described by the author and is discussed in more detail in section 5.2. Thus, one may claim that the reproduced results are more consistent and hence, claim 4 holds as the CE and CR actions are cheaper than ICR.

---

[2]`https://anonymous.4open.science/r/FACT_2024-7C8A/README.md`

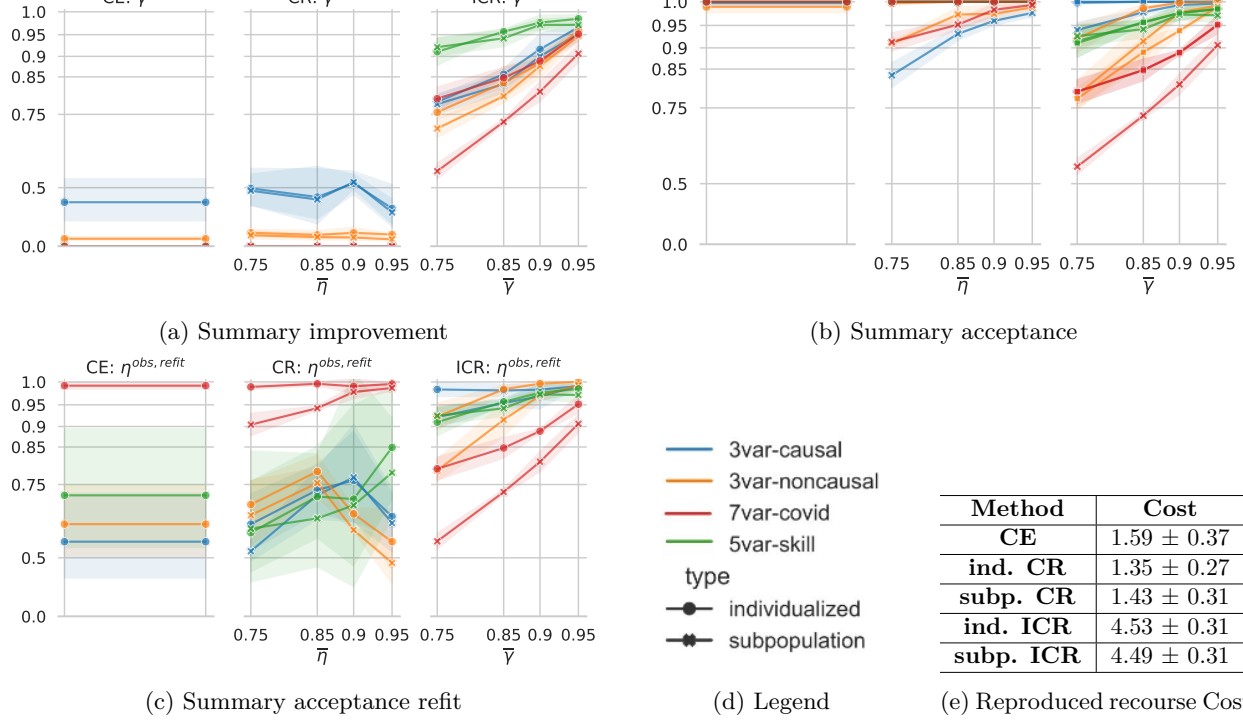

(a) Summary improvement        (b) Summary acceptance

(c) Summary acceptance refit     (d) Legend     (e) Reproduced recourse Cost

| Method | Cost |
|---|---|
| CE | $1.59 \pm 0.37$ |
| ind. CR | $1.35 \pm 0.27$ |
| subp. CR | $1.43 \pm 0.31$ |
| ind. ICR | $4.53 \pm 0.31$ |
| subp. ICR | $4.49 \pm 0.31$ |

Figure 2: Reproduced results for claim 1 - 4

## 4.2 Extended Results

**Ablation - deletion of special edges** We see that if we delete the special edges connecting the exogenous variable with an effect variable, the ICR methods are still better than CE and CR in terms of improvement (Figure 3b) and acceptance(Figure 6a). The results of ICR for 5-var SCM are robust (Figure: 3e) since they almost remain the same before and after edge deletion. Thus, it satisfies claims 1,2 and 3.

**4 var SCM** As part of the results, we see that the results for ICR are better than CE and CR in terms of improvement(Figure 3a) and acceptance(Figure 5a). However, the results are not robust since we see a drop in performance in model acceptance refit values. Thus, it satisfies claims 1 and 2. Claim 3 is not satisfied since results are not robust.

**Hyper Parameter Tuning** As part of the results, we observe that if we change the adaptive threshold values, the results are more or less the same (Figure 3d ). We observe similar trends if we double the population size ( Figure 8). Thus, it satisfies claims 1 and 2.

**Using Different Mutation Methods** Following the experimental phase with these mutation methods, it is noteworthy that although the outcomes in terms of enhancement and acceptance exhibit similarities, the results obtained through the utilization of Gaussian mutation demonstrate greater consistency. The same is seen as in the Figure 9b. Thus, it satisfies claims 1 and 2.

**Using Different Selection Operators** It is crucial to note that while the outcomes in terms of improvement and acceptance share similarities, the results obtained through the application of Non-dominated Sorting Genetic Algorithm-II (NSGA-II) exhibit greater consistency. The results of these approaches are presented in Figure 10. Thus, it satisfies claims 1 and 2.

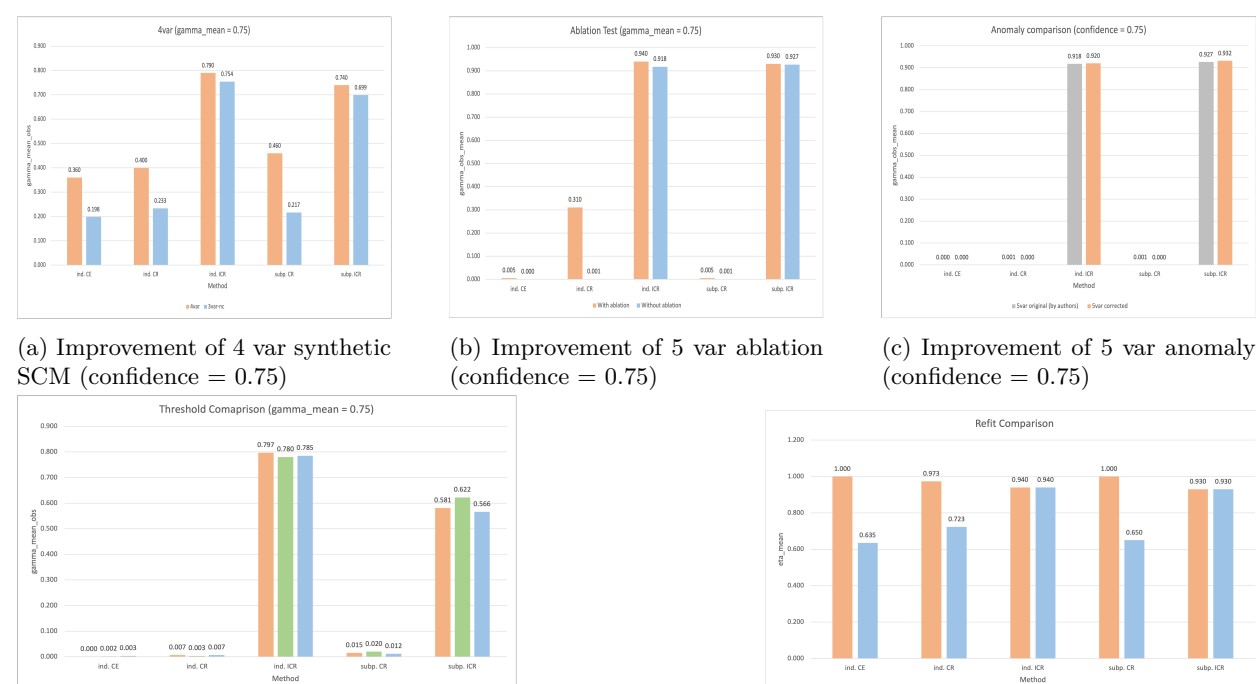

(a) Improvement of 4 var synthetic SCM (confidence = 0.75)

(b) Improvement of 5 var ablation (confidence = 0.75)

(c) Improvement of 5 var anomaly (confidence = 0.75)

(d) Improvement of 7 var threshold test (confidence = 0.75)

(e) Model Refit Acceptance of ablated 5 var (confidence = 0.75) - Tests robustness

Figure 3: Extended results

## 5 Discussion

The extended experiments conducted by us helped in validating the claims by the authors. It proved to be useful in checking how well the code generalises over other settings. An overview of all of these validations can be seen in Table 4.

| | Claim 1 | Claim 2 | Claim 3 | Claim 4 |
|---|---|---|---|---|
| Ablation | confirmed | confirmed | confirmed | not tested |
| Selection | confirmed | confirmed | not tested | not tested |
| Mutation | confirmed | confirmed | not tested | not tested |
| Population size | confirmed | confirmed | not tested | not tested |
| Threshold | confirmed | not confirmed | not tested | not tested |
| 4 var synthetic | confirmed | confirmed | not confirmed | not tested |

Table 4: Overview of proven claims

### 5.1 Easy/difficult aspects during implementation

**Easy Aspects:** Adjusting hyperparameters proved to be a straightforward task due to the original code avoiding extensive hard coding. This flexibility allowed for efficient modifications for our threshold and population size test. The ease of generating plots, due to the well-structured code specifically designed for plot generation, was another advantageous aspect. Furthermore, the synchronisation of variable names between the code and the paper helped in the interpretation process, enhancing overall understandability.

**Difficult Aspects**: The initial challenge we faced was related to the original code's runtime length. It took a significant amount of time to execute as seen in Table 2, showcasing the need for optimisation. Additionally,

the absence of documentation regarding the use of random seeds presented a hurdle, resulting in deviations in standard deviation values. Moreover, the number of iterations in the code was set to 3 whereas in the report, it was stated to be 10. However, the reason for this was clarified later on by the author 5.3. The final issue was once again related to the code. There were some type casting issues with the variables which needed to be fixed before we could start running the code.

## 5.2 Possible bug - 5 var skill SCM

We found a possible bug in the source code for the 5 var skill SCM, where the **languages** variable was replaced with **fever** and **fatigue** variables. We also conducted an experiment where we deleted these 2 variables and added the **languages** variable in the source code and saw that the results have actually improved ( Figure 3c).

## 5.3 Communication with authors

Luckily, the authors had responded to clarify certain areas, which helped clarifying certain design choices and motivations.

The authors stated that the results were not based off of a fixed seed, in order to create diversity for each run. Various outcomes would be produced, because each iteration would use a different seed. This process was essential to bring in randomness. Next, the authors mention that the nr_runs parameter has been set to 10 for each configuration. This is done to maintain the consistency across all the 10 runs. Finally, they addressed the question about the sub-population based Improvement Probability (ICR) method. They clarified that the method does take into account individuals which were accepted pre-recourse. This results in a lower improvement rate due to the presence of a subgroup with both initially accepted and rejected individuals.

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

# A    Original authors results

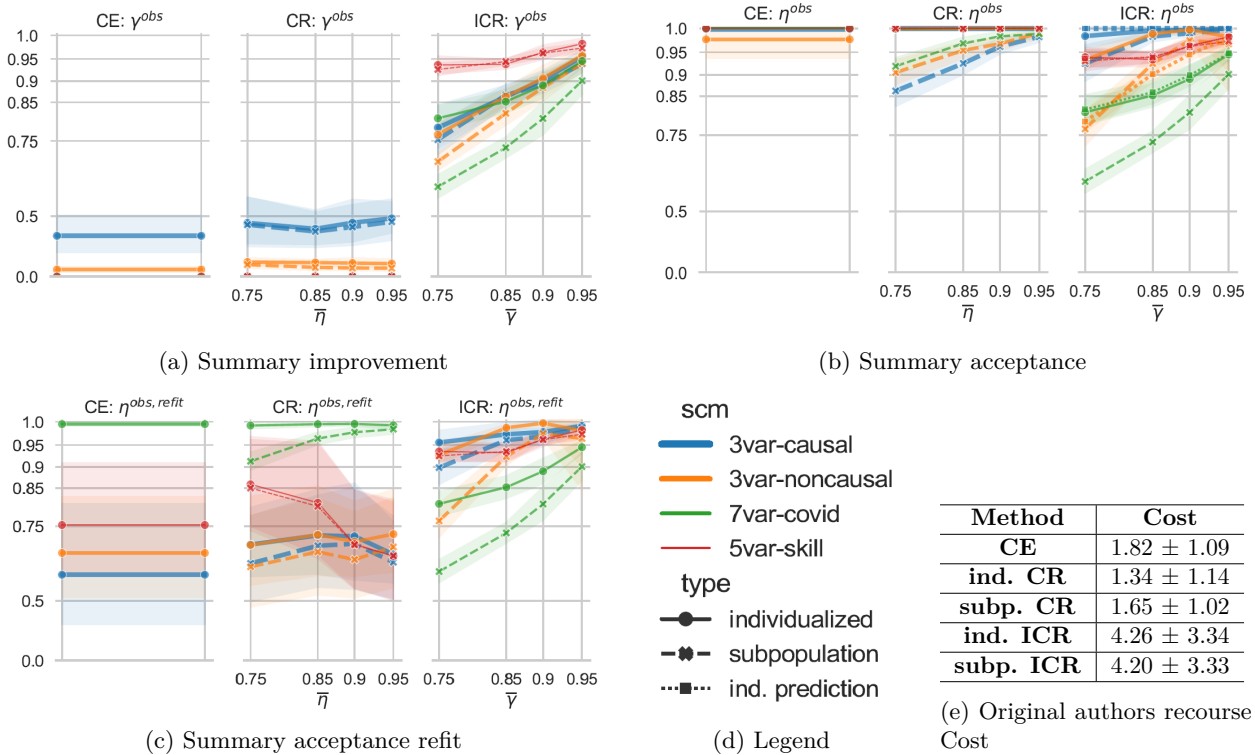

(a) Summary improvement

(b) Summary acceptance

(c) Summary acceptance refit

(d) Legend

| Method | Cost |
|---|---|
| **CE** | $1.82 \pm 1.09$ |
| **ind. CR** | $1.34 \pm 1.14$ |
| **subp. CR** | $1.65 \pm 1.02$ |
| **ind. ICR** | $4.26 \pm 3.34$ |
| **subp. ICR** | $4.20 \pm 3.33$ |

(e) Original authors recourse Cost

Figure 4: Claims 1 - 3

# B    Extended results

This section contains the graphs for the extended results. All the experiments have been run with 0.75 confidence level. We could not run all the extended experiments with every possible confidence level due to computational constraints. Thus, we chose the 0.75 value as it gave us the best results.

## B.1    4 var SCM

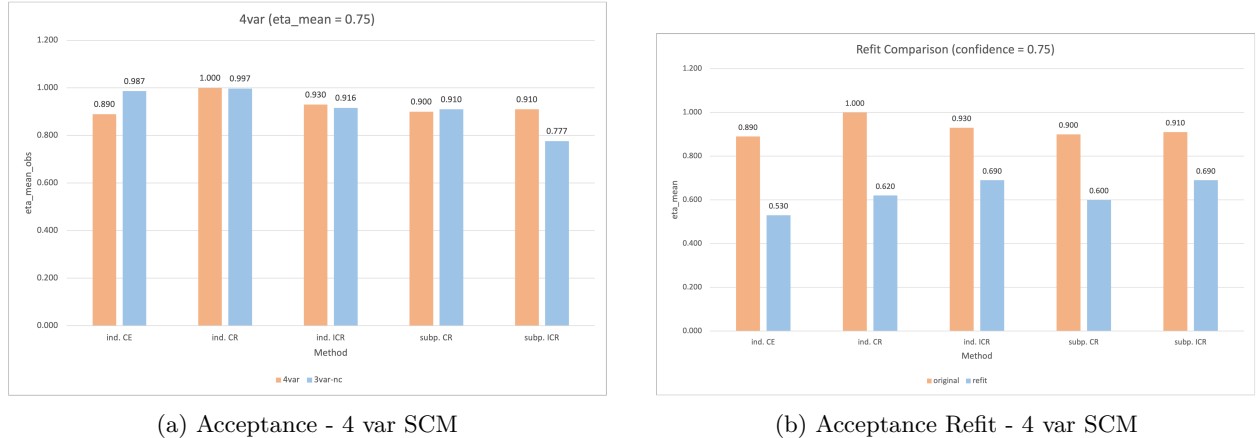

(a) Acceptance - 4 var SCM

(b) Acceptance Refit - 4 var SCM

Figure 5: 4 var SCM

## B.2 Ablation

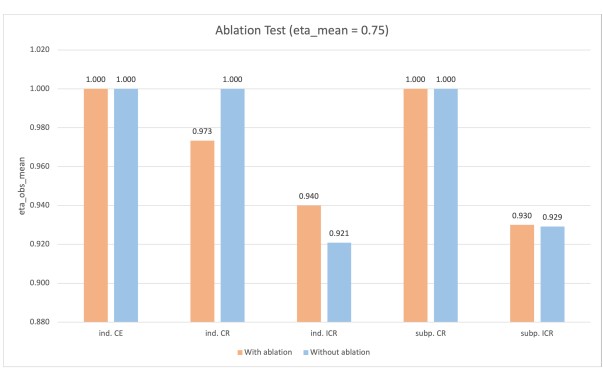
(a) Acceptance - Ablation 5 var

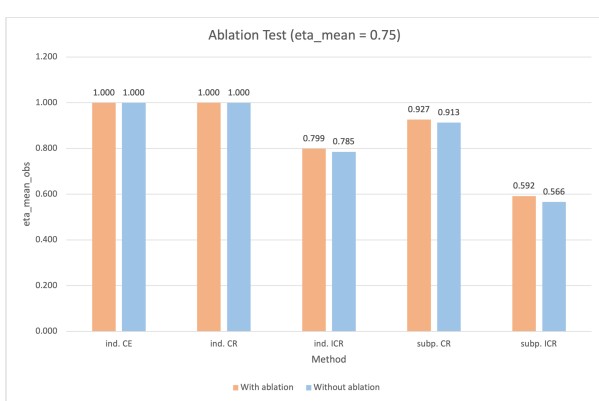
(b) Acceptance - Ablation 7 var

Figure 6: Ablation

## B.3 Anomaly

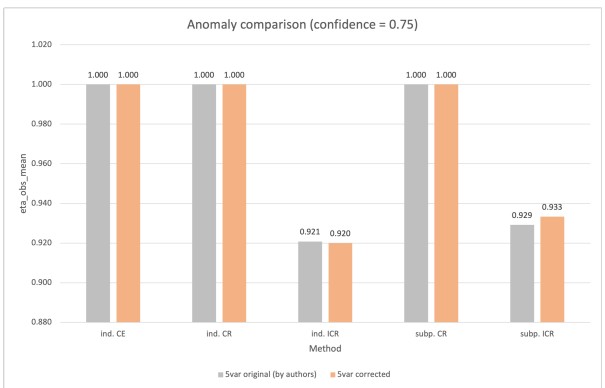

Figure 7: Acceptance - Anomaly 5 var

## B.4 Hyperparameters

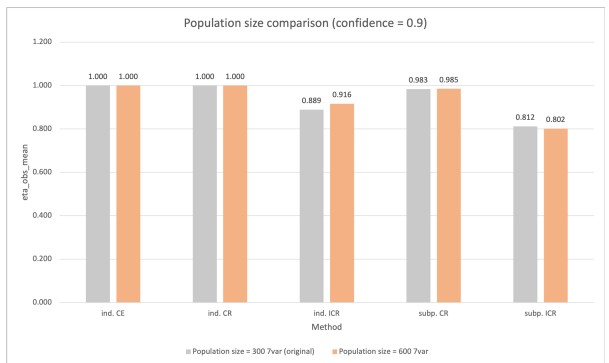
(a) Acceptance - Pop size 7 var

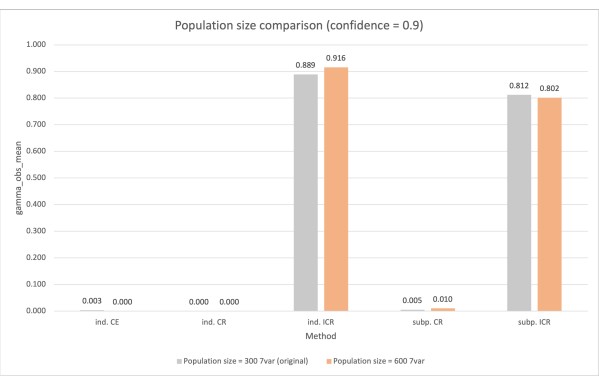
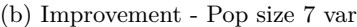
(b) Improvement - Pop size 7 var

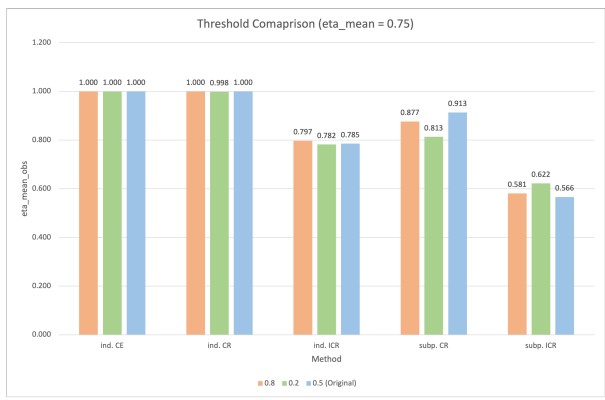
(c) Acceptance - Threshold 7 var

Figure 8: Hyperparameters

## B.5 Mutation

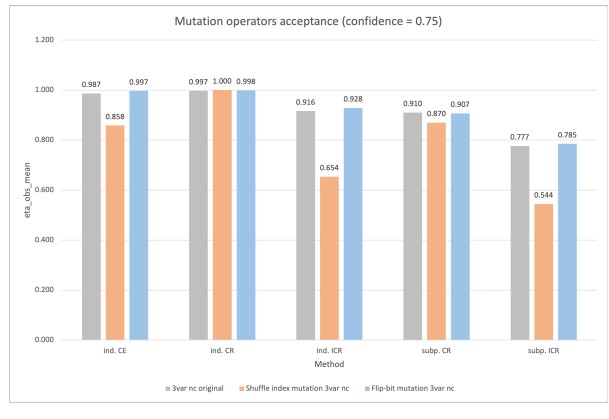
(a) Acceptance - Mutation 3 var nc

(b) Improvement - Mutation 3 var nc

Figure 9: Mutation

## B.6  Selection

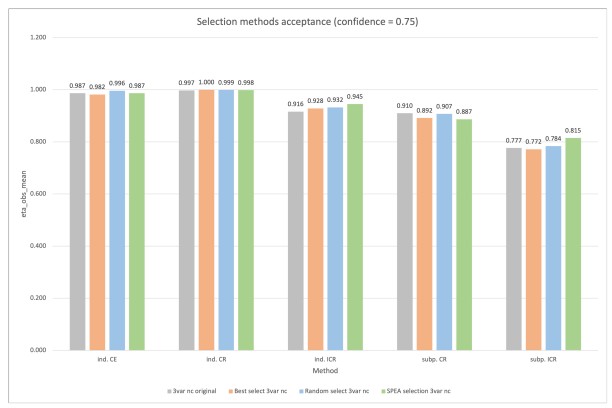

(a) Acceptance - Selection 3 var nc

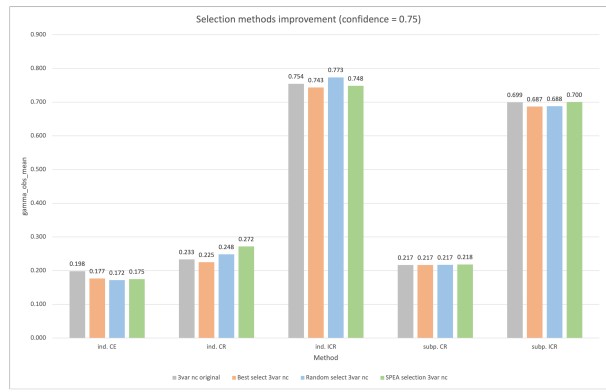

(b) Improvement - Selection 3 var nc

Figure 10: Selection

## B.7  Code anomaly

```python
202 ∨  SCM_PROGRAMMING = GenericSCM(
203         dag=DirectedAcyclicGraph(
204             adjacency_matrix=np.array([[0, 0, 1, 1, 0, 0],
205                                        [0, 0, 1, 0, 0, 0],
206                                        [0, 0, 0, 1, 1, 1],
207                                        [0, 0, 0, 0, 0, 0],
208                                        [0, 0, 0, 0, 0, 0],
209                                        [0, 0, 0, 0, 0, 0]]),
210             var_names=['years_experience', 'degree', 'senior-level_skill', 'nr_commits', 'nr_languages', 'nr_stars']
211         ),
212         noise_dict={'years_experience': dist.GammaPoisson(8, rate=8/3),
213                     'degree': dist.Categorical(probs=np.array([0.4, 0.2, 0.3, 0.1])),
214                     'senior-level_skill': unif_dist,
215                     'nr_commits': dist.GammaPoisson(40, rate=40/4),
216                     'nr_languages': dist.GammaPoisson(2, rate=2/4),
217                     'nr_stars': dist.GammaPoisson(5, rate=5/4)
218                     },
219         fnc_dict={'senior-level_skill': fn_skilled, 'nr_commits': fn_nr_commits, 'nr_stars': fn_nr_stars, 'fever': fn_fever,
220                   'fatigue': fn_fatigue},
221         y_name='senior-level_skill',
222         sigmoidal=['senior-level_skill'],
223         costs=[5.0, 5.0, 0.0001, 0.01, 0.1],
224         bound_dict={'years_experience': (0, sys.maxsize), 'degree': (0, 3),
225                     'nr_commits': (0, sys.maxsize),
226                     'nr_languages': (0, sys.maxsize),
227                     'nr_stars': (0, sys.maxsize)}
228     )
```

Figure 11: Anomaly github

