# OpenReview forum: "Reproducing Improvemement-Focused Causal Recourse"
_TMLR — Rejected by TMLR_

### Review · Reviewer_XDaL · 2024-03-19

**Summary Of Contributions:**

The paper evaluates the reproducibility of König et al.'s (2023) work on Improvement-Focused Causal Recourse (ICR), which addresses concerns about algorithmic fairness and bias in decision-making processes. The study aims to replicate the original paper's four main claims regarding the effectiveness, acceptance rates, stability, and cost considerations of ICR compared to Counterfactual Explanations (CE) and Causal Reasoning (CR). Despite encountering challenges, the replication largely supports the original claims, with some additional investigations conducted to enhance the validity of the findings.

**Audience:**

Yes

**Claims And Evidence:**

Yes

**Requested Changes:**

The work seems to be sufficient for the validation.

**Strengths And Weaknesses:**

[Strengths]

1. The study extends beyond reproducing the original paper's experiments by conducting additional investigations, such as ablation techniques with multiple variables and employing a customized synthetic structural causal model (SCM). Additionally, the authors analyzed the impact of the hyperparameter on the result, and identified a bug in the source code for the 5 var skill SCM.

2. By explicitly outlining the four main claims of the original paper and structuring the reproduction efforts around these claims, the study maintains a clear focus and ensures that the replication process directly addresses the core findings of the original work. This approach enhances the interpretability and relevance of the replication results.

3. The inclusion of an extra experiment to correct issues with the original 5-var dataset highlights the study's commitment to accurate reproducibility and integrity in research. Addressing such discrepancies enhances the reliability of the replication efforts and contributes to the credibility of the findings.

[Weaknesses]

1. This paper presents a concentrated examination of König et al.'s work (2023), dedicating its entirety to the validation of the aforementioned study. However, its focus is strictly limited to this particular work, with no evident attempt to extend its implications or findings beyond the specific context of König et al.'s research. The paper's narrow scope raises questions about its potential impact and relevance to a wider audience. Without a clear connection to broader themes or issues, it may struggle to engage readers outside of those with a specific interest in the validation of König et al.'s work. Consequently, the paper's appeal and significance may be limited, potentially reducing its overall impact in the wider ML community.

2. Although the study utilizes datasets from existing causal studies by Montandon et al. (2021) and Jehi et al. (2020), the reliance on a relatively small set of datasets may lead to overfitting and limited generalizability of the results. The datasets primarily focus on abstract, synthetic settings and specific domains such as developer roles and COVID-19 outcomes. This narrow focus may not capture the full complexity and diversity of real-world scenarios, potentially biasing the results and limiting their applicability to broader contexts. Incorporating a more diverse range of datasets from various domains and contexts could provide a more comprehensive understanding of Improvement-Focused Causal Recourse's effectiveness and mitigate the risk of overfitting, thus enhancing the validity and utility of the findings.

3. Resource Constraints Persist: While the study expands on the original experiments, resource constraints continue to pose challenges, as indicated by the inability to run experiments for 10 iterations as mentioned in the original paper. These limitations may impact the comprehensiveness of the replication effort and potentially introduce biases or limitations in the findings.

---

### Review · Reviewer_oLzz · 2024-03-20

**Summary Of Contributions:**

This is an experimental re-evaluation of the work “Improvement-Focused Causal Recourse (ICR)” König et al. (2023), based on the original code.

**Audience:**

No

**Claims And Evidence:**

Yes

**Requested Changes:**

An independent code implementation is highly desirable if not necessary, and a full re-writing of the paper is needed.

Footnote 1 seems to reveal where the author(s) are located, which is not good.

**Strengths And Weaknesses:**

### Strengths

The effort in the direction of reproducible research is worthwhile.

The paper is quite transparent about what work has been done.

### Weaknesses

The work uses the code from the author(s) of the original work which needs to be reproduced, while I think the most important work in a reproduction is a re-implementation of the original paper, i.e., an independent code project.

It is not clear in the paper why the chosen experiments are important to understand the original work. For example, for edge ablation, the paper says “we try to understand the importance of these edges and their possible effects”, but when the edges are (not) important, what do we say about the original work? For mutation and selection methods, it is not clear why these components are important in the original work.

The following point is related to the question “*Would at least some individuals in TMLR's audience be interested in knowing the findings of this paper?*”. It seems to me the original work is an incremental work on Causal Reasoning (CR) Karimi et al. (2020) and it is not well cited even considering its age (10 cites on Google Scholar, at least 4 of them are from the research group of the original work). Thus, even if the author(s) address the two weaknesses above, I am not sure if there are a hundred persons who will be interested (this is my interpretation of *some individuals* in the current context).

---

### Review · Reviewer_jraa · 2024-03-27

**Summary Of Contributions:**

The authors reproduce experiments from a paper about improvement-focused causal recourse.

**Audience:**

No

**Claims And Evidence:**

No

**Requested Changes:**

*Critical*:
- elaborate and make more explicit what this submission adds beyond reproducing results of the unpublished paper referenced in the last sentence of the introduction (and also clarify how that paper differs from the one published in 2023)
- Discuss claims more carefully: the claims as stated in Section 2 are fundamentally not the sort that can be proven by running a finite number of experiments. The claims should be amended to be about specific data sets (so that they can be proven by a finite number of experiments), or they supported with actual mathematical proofs, or the submission should consistently use words like "supported" and "corroborated" instead of "confirmed" and "proven"
- provide more references, definitions, and careful presentation of the ideas in the paper, because this should still be a stand-alone submission that properly introduces and references things like SCMs and the do operator (and e.g., the "4 var SCM" has 5 variables, and its description confuses exogenous and endogenous variables, among other things)
- fix formatting (e.g., use parenthetical vs in-text citation correctly, make text inside plots big enough to be legible, carefully check fonts in math environments, ...)

*Suggested*:
- offer explanation for why these experiments take so long to run even though the graphs are quite small and the experiments are run on a super computer

**Strengths And Weaknesses:**

The main strength is that it provides modest insights about experiments and claims of a previous paper.

The main weaknesses is that it's not clear what else is added beyond reproducing the previous paper's results (and the previous paper itself seems to have had only a modest impact).

---

### Review · Reviewer_Bk3k · 2024-03-27

**Summary Of Contributions:**

The paper examines the reproducibility of the original work by König et al. (2023) regarding Improvement-Focused Causal Recourse (ICR). The authors aim to validate several claims made by the original paper, including ICR's superiority in terms of improvement, its ability to achieve desirable acceptance rates, the stability of these rates across model refits, and the cost comparison between ICR, CE, and CR interventions. Through a series of experiments replicating the original study and introducing extensions for further validation, the paper confirms most of the claims tested across various synthetic and real-world datasets.

**Audience:**

Yes

**Broader Impact Concerns:**

None.

**Claims And Evidence:**

Yes

**Requested Changes:**

Expand the discussion on how ICR can be applied in real-world settings, its potential benefits, and challenges in practical implementations.

**Strengths And Weaknesses:**

Strengths:

-	The paper thoroughly evaluates the reproducibility of the original study's claims across different datasets and experimental settings, providing a robust assessment of ICR's effectiveness.

-	By extending the research through additional experiments, such as ablation studies and hyperparameter tuning, the paper contributes to a deeper understanding of the ICR model's generalizability and limitations.

Weaknesses:

-	The analysis of cost implications is somewhat limited to the computational aspect, with less emphasis on the practical costs associated with implementing the suggested interventions.

-	While the paper provides a thorough comparison with the original study, it could include a broader review of recent literature to position its findings within the current research landscape.

---

### Decision · Action_Editor_VZMg · 2024-05-10

**Recommendation:** Reject

**Comment:**

Introducing a wider range of datasets, and making a connection between this reproducibility study and recent research would help make this work more impactful.

**Audience:**

Multiple reviewers suggested a more thorough review of recent literature to better position its findings and claims among current research.  Understanding this context would bolster the argument that there is indeed an audience for this work.  In its absence, it is hard to make the argument that this reproducibility study will find much of an audience.

**Claims And Evidence:**

Reviewers generally agreed that this reproducibility submission did not yield much insight beyond the original paper.  Specifically, claims made in section 2 are extremely broad and presented without caveats.  As reviewer jraa stated: “the claims as stated in Section 2 are fundamentally not the sort that can be proven by running a finite number of experiments”.  Reviewer XDaL comments, “the reliance on a relatively small set of datasets may lead to overfitting and limited generalizability of the results.”  Another reviewer notes that the reproduction was done with code provided by the original authors.  While vetting the original code has value, re-running it risks committing correlated errors, and limiting understanding of the underlying methodology as opposed to idiosyncrasies of a particular implementation.

**Resubmission Of Major Revision:**

The authors may consider submitting a major revision at a later time.